# Melt probabilities and surface temperature trends on the Greenland ice sheet using a Gaussian mixture model

Daniel Clarkson[1], Emma Eastoe[1], and Amber Leeson[2]

[1]Department of Mathematics and Statistics, Lancaster University, Lancaster, United Kingdom.
[2]Data Science Institute, Lancaster University, Lancaster, United Kingdom.

**Correspondence:** Daniel Clarkson (d.clarkson@lancaster.ac.uk)

**Abstract.** The Greenland ice sheet has experienced significant melt over the past six decades, with extreme melt events covering large areas of the ice sheet. Melt events are typically analysed using summary statistics, but the nature and characteristics of the events themselves are less frequently analysed. Our work examines melt events from a statistical perspective by modelling 19 years of Moderate Resolution Imaging Spectroradiometer (MODIS) ice surface temperature data using a Gaussian mixture model. We use a mixture model with separate model components for ice and meltwater temperatures at 1139 cells spaced across the ice sheet. By considering the uncertainty of the ice surface temperature measurements, we use the two categories of model components to define, for each observation, a probability of melt which is independent of any pre-defined fixed melt threshold. This probability can then be used to estimate the expected number of melt events at a given cell. Furthermore, the model can be used to estimate temperature quantiles at a given cell, and analyse temperature and melt trends over time by fitting the model to subsets of time. Fitting the model to data from 2001-2009 and 2010-2019 shows increases in melt probability and yearly expected maximum temperatures for significant portions of the ice sheet.

## 1 Introduction

The Greenland ice sheet has experienced significant melt over the past six decades (Fettweis et al., 2011) and has had an overall accelerating contribution to sea-level rise from a combination of melt and dynamical discharge, in particular over the last 18 years (Rignot et al., 2018). Wide regions of the ice sheet have lost mass over the last two decades resulting in an increasing contribution to sea-level rise (Mouginot et al., 2019). Combined with melt from other ice bodies, e.g. the Antarctic ice sheet and valley glaciers, groundwater depletion, and thermal expansion of the oceans, total sea-level rise has been far above the historical rate of sea-level rise during this period (Chen et al., 2017). Understanding where, when, and how frequently melt occurs on the Greenland ice sheet is a key part of understanding its role in sea-level rise and how we might expect it to change in the future.

Since air temperature is a strong control on ice melt (Vermeer and Rahmstorf, 2009), temperature data is often used as a proxy for melting. Ice surface temperatures exceeding $-1°C$ can be interpreted as evidence of melt depending on the dataset used and its accuracy and uncertainty (Hall et al., 2018). There are many ways to study the temperature of the ice sheet, including through observations from space (Zhengming and Dozier, 1989), Automatic Weather Stations (AWSs) (Tedesco

et al., 2013), and using Regional or Global Climate Model output (Smith et al., 2007). Data from these diverse sources are characterised by differing levels of accuracy and coverage. Whilst in-situ observations are often considered to provide the most accurate measurements for a given cell, and Global Climate Model (GCM) output allows consideration of temperatures under different climate scenarios, satellite data has comparable accuracy to in-situ measurements under clear-sky conditions (Hall et al., 2008) but with far higher spatial coverage, thus providing the most comprehensive overall view of the ice sheet.

In 2012, a record-breaking melt event was observed during mid-July, with $98.6\%$ of the ice-sheet simultaneously experiencing melt (Hanna et al., 2014). Extreme melt events such as this are likely to become more common as overall temperatures on the Greenland ice sheet increase, contributing to increasing amounts of melt. Despite their contribution to our overall understanding of melt on the ice sheet, the magnitude, frequency, and melt contribution of these melt events are not clearly defined. Because these events are rare, our understanding of them has necessarily been based on case studies of a few isolated examples to date. By applying statistical models to these events, we can both deepen our understanding of the physical properties of the melt events and improve quantification of melt on the ice sheet overall.

Here, we propose a novel statistical approach applied to Moderate Resolution Imaging Spectroradiometer (MODIS) Ice Surface Temperature (IST) data to model the distribution of temperatures on the Greenland ice sheet at 1139 MODIS cells, with a particular interest in identifying and modelling melt temperatures. The approach is based on three key characteristics of IST data: firstly, the presence of physical bounds on the range of ice and ice-melt temperatures; secondly, the multi-modality of the distribution; and thirdly, ambiguity about whether measurements close to $0^{\circ}$C represent melting of the ice sheet surface. This model based approach has several advantages over a purely empirical analysis, including allowing full characterisation of the distribution of ist and resulting properties e.g. melt threshold exceedance probabilities, quantiles, return periods and return levels, as well as allowing for out-of-sample prediction and extrapolation. Since the sample of cells used to fit the statistical model is uniformally distributed over the full ice sheet, our model is sufficiently general-isable as to be useful for cells not explicitly used to generate the model, regardless of elevation, distance from the coast, or geographical location. Finally note that we limit our analysis to the modelling of cloud-free days. This is due to the absence of data on days with cloud cover and the bias that would ensue if we were to assume that temperatures on clear days could be used to represent these missing values. The data can not be considered as missing at random, so there would be a bias in temperatures on cloudy days compared to clear days. We use this model to investigate time trends in the observation period and to quantify both the frequency and magnitude of temperature events that are likely to result in ice melt.

## 2 Data and methods

### 2.1 MODIS IST data

We use MODIS IST data from MODIS/Terra Sea Ice Extent 5-Min L2 Swath 1km, Version 6 (MOD29) contained within a multilayer Greenland MODIS-based product (Hall et al., 2018). MODIS records surface reflectance from 36 spectral bands of different wavelengths - including those used in IST - near daily for the entire Earth. This dataset spans the period 01/03/2000 to 31/12/2019 and has a spatial resolution of .78 km $\times$ .78 km. Here we discard the first 10 months of the data set, up to

01/01/2001, in order to work only with those years for which a full annual cycle is available. To reduce the computational burden of our model, we also subsample the data taking 1 in every 50 cells in both $x$ and $y$ dimensions for a total of 1139 cells,

roughly equally spaced across in latitude and longitude and thus covering the full range of glaciological and climatological settings across the ice sheet.

## 2.2    Cloud cover

The IST measurements represent the temperature at the surface of the ice in cloud-free conditions. Clouds (specifically water vapour) can interfere with the measurements, so a cloud mask is used in the MODIS product to remove measurements made

in cloudy conditions. As a consequence, our analysis and predictions are valid for clear conditions only. Due to the generally warmer temperatures seen on cloudy days, were the analysis to be interpreted as representative of clear days also, there would be a strong likelihood of over over-estimating the magnitude and frequency of melt events (Koenig and Hall, 2010).

As a result of cloud masking, areas on the coast and in the north have a higher proportion of missing data than more central areas (Figure 1). We also see that winter months have more missing data on average than summer months because of cloud

cover, with a range of $65.1\%$ of data available in December compared to $91.1\%$ of data available in May. This is important to bear in mind when interpreting the predictions made from the statistical models, as the IST distributions will be more heavily weighted towards warmer temperatures. This shouldn't affect our inference with regards to melt, however, as melt temperatures almost exclusively occur in the summer months which have a much lower proportion of missing data.

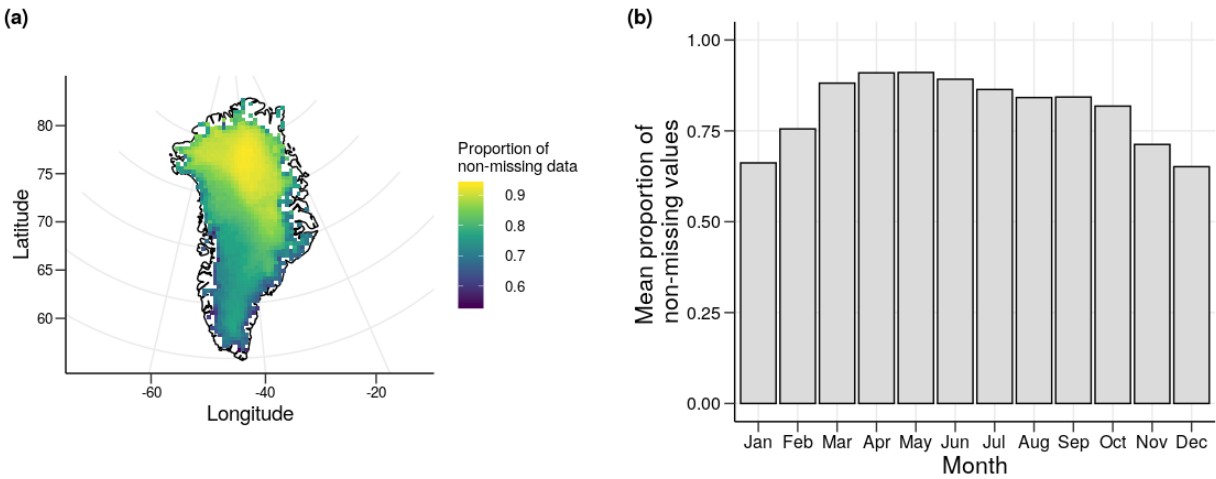

**Figure 1.** Proportion of available MODIS IST data (i.e not filtered by the cloud mask) at 1139 cells on the Greenland ice sheet between 2001 and 2019 (a) and mean proportion of available MODIS IST data by month between 2001 to 2019 (b).

To check for the possible presence of longer term changes in surface conditions in response to changes in cloud cover, the

proportion of data missing due to cloudiness is compared between decades (Figure 2). The clear spatial trends seen in the overall proportions of missing data are not present here, though locations in the north and south of the ice sheet have slightly

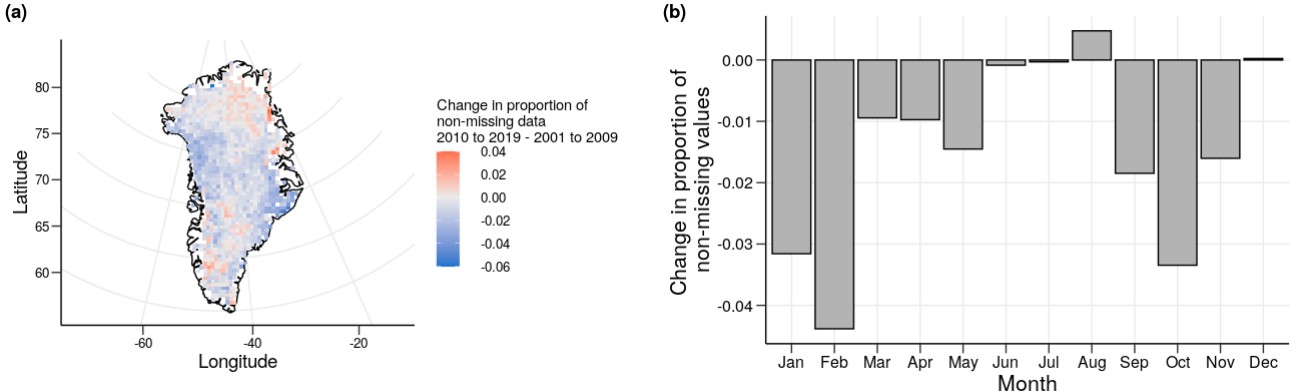

**Figure 2.** Change in proportion of available MODIS IST data (i.e not filtered by the cloud mask) at 1139 cells on the Greenland ice sheet from 2010 to 2019 - 2001 to 2009 (a) and change in mean proportion of available MODIS IST data by month from 2010 to 2019 - 2001 to 2009 (b).

more data in the most recent decade in contrast to locations at mid-latitudes where the reverse is true. However, these trends are not consistent over all locations, and the magnitudes of the changes are relatively minor with a maximum absolute change of 0.060 and a mean absolute change of 0.010. Seasonal differences in cloud cover and thereby data availability also vary

between the two decades. January, February, and October show the largest changes, with a maximum absolute change of 0.044 in February. Ten months have an average decrease in data availability between the decades but the absolute differences are reasonably minor. Furthermore, of the four months with the lowest changes, three (June, July, and August) also see the highest average temperatures, giving further confidence in our cross-decade comparison of temperature quantiles and melt estimates.

## 2.3 Modelling considerations

To create a statistical model that is parsimonious and applicable at all cells over a large and geographically-varied region, we model the IST data using statistical methods that allow us to treat melting in a probabilistic manner. Exploratory data analysis shows that there is no clear quantile in the temperature distribution that can be attributed as the onset of melting (Figure 3). As a result, we model melting ice temperatures and non-melting ice temperatures separately and estimate the probability of melt occurring over a range of temperatures. This approach allows for some uncertainty in the observations from factors such as the

precision of the dataset, which has a stated uncertainty of $\pm 1°$C. We hereby refer to temperatures associated with melting ice as "melt" temperatures and temperatures associated with non-melting ice as "ice" temperatures.

A key feature of the dataset and a core modelling consideration is the soft upper limit at $0°$C. The melting point of the ice acts as a physical upper limit on ISTs, as once the ice exceeds this temperature it melts and may no longer form the surface of the ice sheet. Some sites have measurements above this limit, which arise due to meltwater sitting on top of the ice. However,

the ice under the water places a limit on these melt temperatures, hence the distribution of positive temperatures is truncated

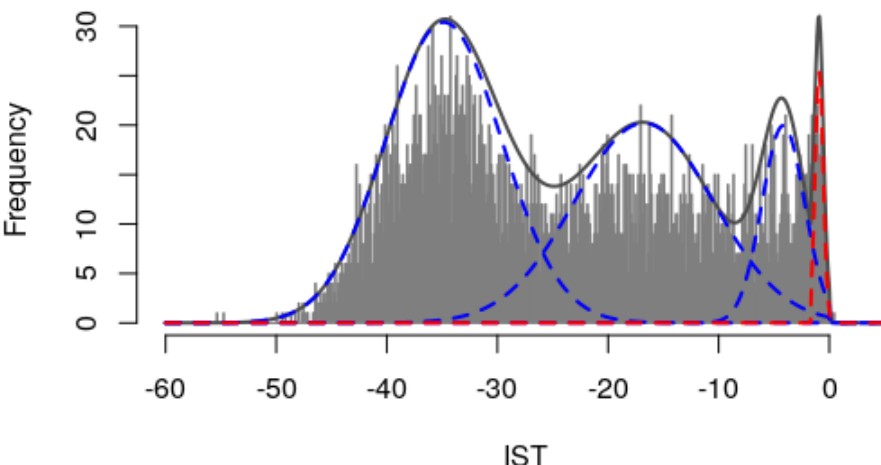

**Figure 3.** Frequency distribution of daily MODIS IST data from an example cell (82.47, -37.50) on the Greenland ice sheet between 2001 and 2019. Solid lines show a mixture model fit to these data where blue indicates the three 'ice' components, red indicates the 'melt' component, and black indicates the full model as the sum of the ice and melt components. Each individual component is a truncated Gaussian distribution, and the lines represent the probability density function of these on a scale to matching that of the histogram.

close to $0°$C. This soft upper limit of ISTs causes a significant peak in the distribution centred at approximately $-0.5°$C, as any ISTs that would exceed $0°$C are truncated to small positive values close to $0°$C.

The simplest statistical model would be to fit a single distribution to the full data set, potentially after an initial transformation. This raises two issues. Firstly, a bimodal distribution is clear at all cells with cells that experience melt having the highest mode close to $0°$C (Figure 3). For non-melt cells, the location of the higher mode is more variable. This does not appear to be directly attributable to seasonal differences in temperatures, as as the shape of seasonal temperature distributions show as much inter-site variability as they do inter-season variability. Fits of unimodal distributions are particularly poor at the tails of the distributions, which is particularly problematic since our interest lies in melt which is directly connected to the upper tail of the temperature distribution.

Given the focus on melt, an alternative option would be to undertake an extreme value analysis of only the highest temperatures at each cell. This would allow the model to focus on the temperatures of highest interest that are the most difficult for more standard models to capture. This also proves problematic though, as in order to fit the model the temperatures must first be identified as extreme using an extreme value threshold, with temperatures above the threshold being classed as extreme and those below being classed as non-extreme. Due to the mode around $0°$C, finding a consistent threshold location using quantiles, gradient analysis or a specific temperature all encounter problems due to the large variety of tail shapes at different cells. Each

of the above mentioned threshold types do not work universally across the ice sheet, and in many cases provide much worse fits than a distribution applied to the whole range of temperatures.

A consistent model that can be automatically applied at cells across the ice sheet therefore requires a multi-modal distribution or a time-series model to capture seasonal behaviour. The disadvantage of the latter is that it is less able to capture the mode around $0°C$ and the truncation of the temperatures which is where our research interest lies. This further motivates a modelling approach that more directly considers the distribution of the specific data set and allows for multi-modal distributions. The overall distribution shape is broadly similar between sites with the main difference being the proximity of the distribution to $0°C$ and thereby the amount of truncation in the data. The considerations above around the multi-modality of the data set and of the nature of melt temperatures around $0°C$ give us a basic set of assumptions to base our modelling around that allow the model to retain the same underlying structure regardless of the absolute difference in ISTs between cells.

## 2.4  Model description

In order to accommodate spatial variability in the temperature distribution, we model IST using a truncated Gaussian mixture model in which components are assigned to model groups of temperatures that we assign to be either ice or melt. For $n_I$ ice components and a single melt component, let $\phi_i$ be the weight associated with model component $i$ such that for $n_c = n_I + 1$ total components, $\sum_{i=1}^{n_c} \phi_i = 1$. For each ice component $i$ (and melt component $M$) let $f_i(x)$ be the probability density function of the truncated normal distribution $X \sim TN(\mu_i, \sigma_i^2, a_i, b_i)$, where $\mu_i$ is the mean, $\sigma_i$ is the standard deviation, $a_i$ ($b_i$) is the lower (upper) truncation point. Then the probability density function of ISTs $x$ is:

$$p(x) = \sum_{i=1}^{n_I} \phi_i f_i(x) + \phi_M f_M(x).$$

We set the upper and lower truncation points for the ice and melt components at values that bound each measurement type with relative certainty. For the ice components, $a = -\infty$ as there is no hard lower limit on the temperature of ice (aside from absolute zero), and $b = 0$ as, theoretically, ice temperature can't exceed $0°C$. This means that there is no limit on how low ice temperatures can go, but they can't exceed $0°C$. For the melt component, $b = \infty$ and $a = -1.65$, so that temperatures in the melt component can't go below $-1.65°C$ but are not upper truncated. We take a bound lower than zero here to account for uncertainty in the data and any potential impurities in the ice surface. $-1.65°C$ is the theoretical minimum temperature at which saline ice can melt (Hall et al., 2004), and thus should be a conservative estimate for this lower bound. Temperatures between $-1.65°C$ and $0°C$ can be modelled by either/both the ice and melt components as there is uncertainty as to whether they are associated with melting or non-melting ice.

A mixture model was fitted using the Expectation-Maximisation (EM) algorithm for each sample cell. The algorithm alternates between two main steps: calculating the component probabilities that each observation $x_i$ comes from model component $k$, and maximising the expectations of the model parameters using the component probabilities (for full details see Appendix A). We used this method to obtain estimates of $\mu$, $\sigma$, and $\phi$ for each model component at each cell.

We used Bayesian Information Criterion (BIC) to assess the most appropriate number of ice and melt components and found that three ice components and one melt component fit the data best. These components may be broadly interpreted as winter, autumn, spring, and the melt season for the three ice components and single melt component respectively.

When modelled with separate Gaussian components, the characteristics of the different modes of the data are much clearer (Figure 3). The melt component at each cell generally has a much lower variance than the ice components due to the soft upper limit of ISTs and the lower truncation point of the model, whereas the ice components have higher variances and more overlap between components. For the sites that experience melt regularly, a substantial proportion of the overall temperature distribution occurs in the overlap between true ice and true melt. A similar result is seen across sites located on or near the coasts, which further validates the decision to use a fixed melt threshold as the melt temperatures - and thereby the melt process - appear to have consistent characteristics across cells.

## 2.5 Defining melt

Using this model, the probability of melt occurring, which we denote by $\rho(x)$, can be quantified as the ratio of the densities of the ice and melt components. For a given IST $x$, $n_I$ ice components melt component $M$, we have:

$$\rho(x) = \frac{f_M(x)}{f_M(x) + \sum_{i=j}^{n_I} f_i(x)}.$$

Consequences of this definition are that for ISTs below $-1.65°$C, the probability of melt is 0, for ISTs above $0°$C the probability of melt is 1, and between these values the melt probability depends on relative values of the melt and ice components' densities. For cells with very few or no ISTs above $-1.65°$C, the weight of the melt component may be close to or equal to 0, in which case the probability of melt occurring is effectively zero. Note that there are discontinuities in the model-based estimate of this probability due to the censoring of the mixture components. These discontinuities occur at the edges of the range of interest ($-1.65°$C and $0°$C) and are more or less severe depending on the degree of truncation of the ice and melt components.

## 3 Results

### 3.1 Melt extent comparison

Using our model, we calculate the expected number of melt days in each year at each sample cell. Let $N_y$ be the number of melt days in year $y$, then $\mathrm{E}[N_y] = \sum_{i=1}^{m} \rho(x_i)$ where $\rho(x)$ is the notation introduced earlier to denote $\Pr[\text{melt}|X = x]$ and $m$ is the number of observations in year $y$. The overall annual average is simply the average of the individual annual averages. We then compare our modelled estimates to a simple threshold-based approach to defining melt, i.e. the average number of days per year with temperatures greater than or equal to $-1°$C (Figure 5).

The majority of the ice sheet - 90.7% of cells from the expected melt from the model, 79.5% from a threshold of $-1°$C on the data - experiences some degree of melt on average each year, except for sites in the dry snow zone in the centre and north of the ice sheet (Benson, 1960). Of the cells that experience melt, most (62.2% from the model, 57.3% from the data) sites on

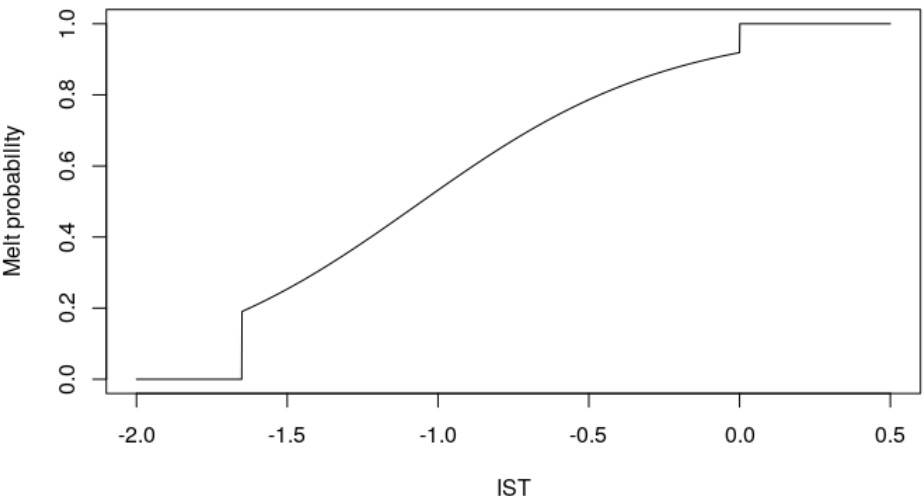

**Figure 4.** Melt probability estimates of a range of ISTs using the fitted mixture model at a single cell (75.37,-58.13) on the Greenland ice sheet between 2001 and 2019. Because some cells have very limited data above $-1.65°$C, we use a cell on the west coast with a high proportion of data above $-1.65°$C (22.55%), thus giving us an increased amount of information in the most pertinent temperature range.

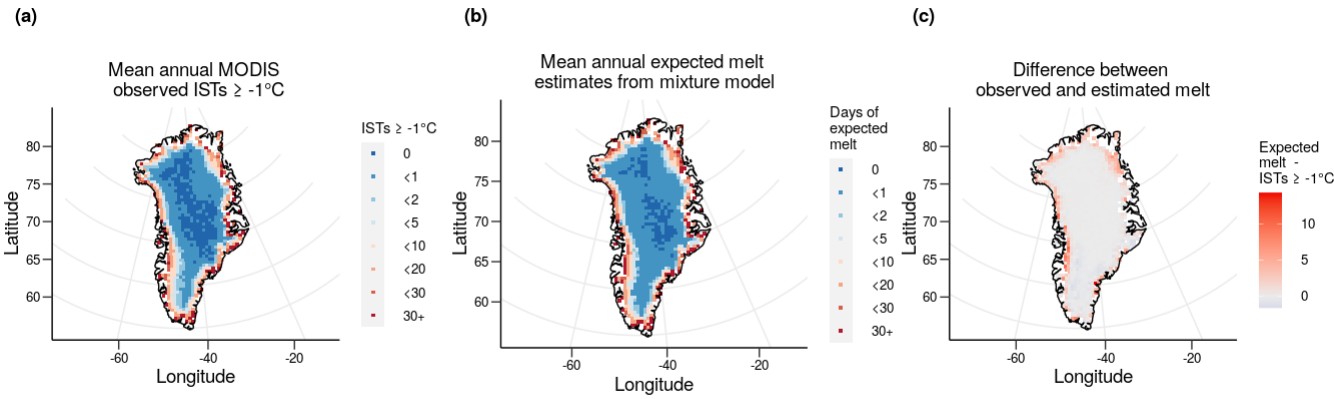

**Figure 5.** The mean annual number of ISTs above $-1°$C per year (a), the mean annual expected melt days estimated from each cell's mixture model (b), and the difference between the two variables (c).

average see less than 2 days of melt per year, which makes up the rest of the dry snow zone and most of the percolation zone. The areas with the most melt are located around the coast and in the south and west as may be expected. The main discrepancies between the two measures are at coastal cells, particularly on the west and north coasts. Here, the model estimates a larger

amount of melt, with a maximum of 14 additional melt days at 1 specific cell on the edge of the south east coast compared to the dataset. However, 89% of cells have an absolute difference of less than 2 melt days, showing the broad agreement between the measures at central cells.

## 3.2   Comparison to AWS melt statistics

To place the methodology and the results within the context of the literature, we compare the estimates of melt found using
our method to those estimated using data from the Programme for Monitoring of the Greenland Ice Sheet (PROMICE) AWSs (Fausto et al., 2021). The characteristics of the datasets differ with known biases between the two (Koenig and Hall, 2010), however previous validation carried out independently for both datasets suggests it is reasonable to consider both datasets as representative of the true surface temperatures. With AWS as a baseline, Figure 6 compares the difference between the AWS melt proportions with the same quantity obtained from (a) MODIS IST and (b) the model fitted to MODIS IST. Despite
the relatively large distances between some of the comparison locations, there is still reasonable agreement for a substantial number of AWS locations. The differences between the estimates may also be partially due to other factors surrounding the nature of the comparison, such as the reduced amount of data available because of the need for common dates, the difference in time resolutions between the datasets in terms of the number of observations averaged to give a single daily observation, and the different measurement uncertainties that lead to different definitions of melt.

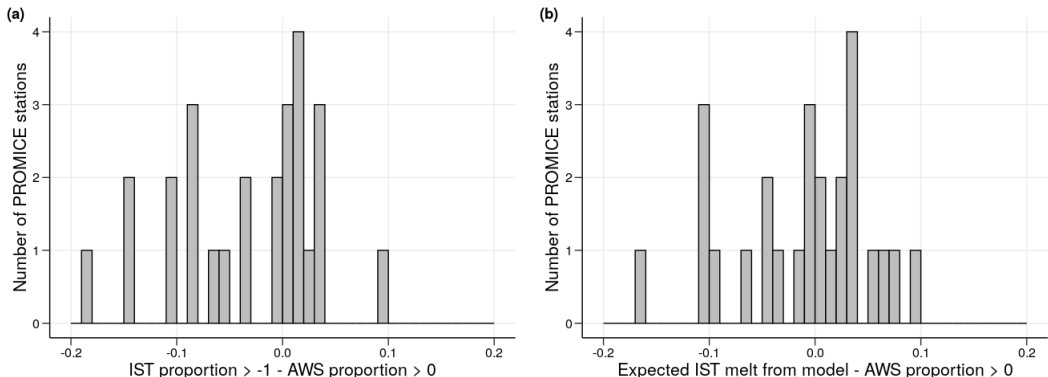

**Figure 6.** A comparison of the empirical melt from PROMICE AWS data (proportion of data $\geq 0^\circ$ C) with empirical MODIS IST estimates (proportion of data $\geq 0^\circ$ C) and mixture model based expected melt estimates. Only dates with data available from both datasets are used, meaning the above estimates are valid for cloud-free days only due to the limitations of the MODIS IST dataset.

## 3.3   Temperature quantiles

We now use the model fit to calculate quantiles of the ISTs at each cell (Figure 5). This gives context to the overall temperature trends observed in the dataset, before looking at melt in more detail. We calculate the 90% quantiles to examine the broader trends of high temperatures that aren't necessarily melt temperatures, as well as the 10% quantiles for temperatures that are as

relatively low as the 90% quantiles are high. The estimated 10% and 90% quantiles broadly follow the same trends as elevation on the ice sheet. The 10% quantiles have a range from $-53.84°C$ in the centre of the ice sheet to a maximum of $-15.75°C$ at the south tip of the ice sheet. As would be expected, cells at higher elevations have a lower 10% and 90% quantile. However, of more interest are the few (30/1139) cells located on the west, east and southern coasts that have a 90% quantile above $0°C$. At these cells, we would expect at least (in some cases more than) 10% of observed temperatures to be above $0°C$ and thereby melt temperatures.

We also calculate the 1-year return levels of each cell. This is the IST that is on average only exceeded once per year as estimated from each cell's mixture model. The return levels range from a minimum of $-7.08°C$ in the centre to a maximum of $7.24°C$ on the west coast. Although, as previously discussed, ISTs should not be seen higher than $0°C$, these return levels reflect similar temperatures recorded by observations in the dataset and can be plausible temperatures when considering the effect that meltwater on the surface of the ice sheet has on the observations. The rarity of melt in certain central areas can be seen more clearly, as temperatures in many cells (519/1139) on average reach $-1.65°C$ less than once a year. The trends seen in the return levels also broadly agree with those seen in the quantiles, and are in reasonable agreement with the elevations and distance to the coast of each cell, with cells at lower altitudes and closer to the coast generally experiencing more melt.

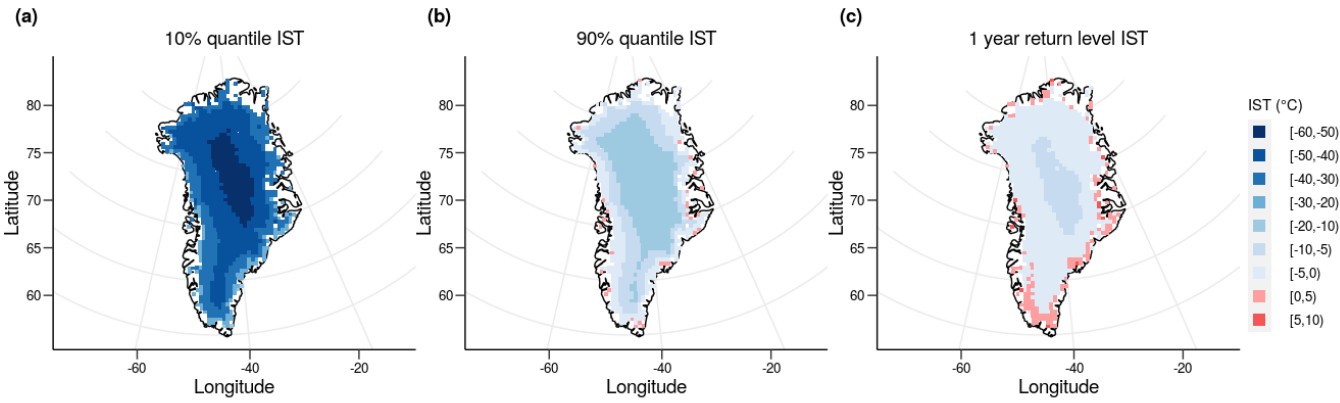

**Figure 7.** 10% quantile (a), 90% quantile (b), and 1 year return level (c) estimates for MODIS IST data from 2001 to 2019 at 1139 cells. Estimates are calculated from fitted mixture models at each cell.

## 3.4 Decadal variability

To examine potential changes in melt over time, we fit mixture models at each cell for two separate decades: 2001 to 2009 and 2010 to 2019. Averaging over a decade helps to smooth some of the annual variability and thus highlight any potential differences as a result of climatic change. To assess any changes in melt and high ISTs between the decades, we compare quantiles between the fitted models and the estimated melt probabilities at each cell in each decade.

### 3.4.1 Temperature quantiles

Because some central areas of the ice sheet do not have many historical melt observations, we examine the 95% quantiles and
yearly expected maximum temperatures, both of which give an indication of overall trends in high temperatures even if these
do not reach the level required for melt at some cells. As previously, we take the estimated quantiles from each of the fitted
decadal models for each cell.

For almost all cells (1100/1139), the 95% quantile increased between the two decades. Cells in the south in particular have
increased fairly consistently. The average change for all cells south of 73.41°N was 0.73°C, with 99.3% of all cells further
south than this seeing an increase. The largest increases were also concentrated in the southern areas of the ice sheet, with a
maximum increase of 1.78°C.

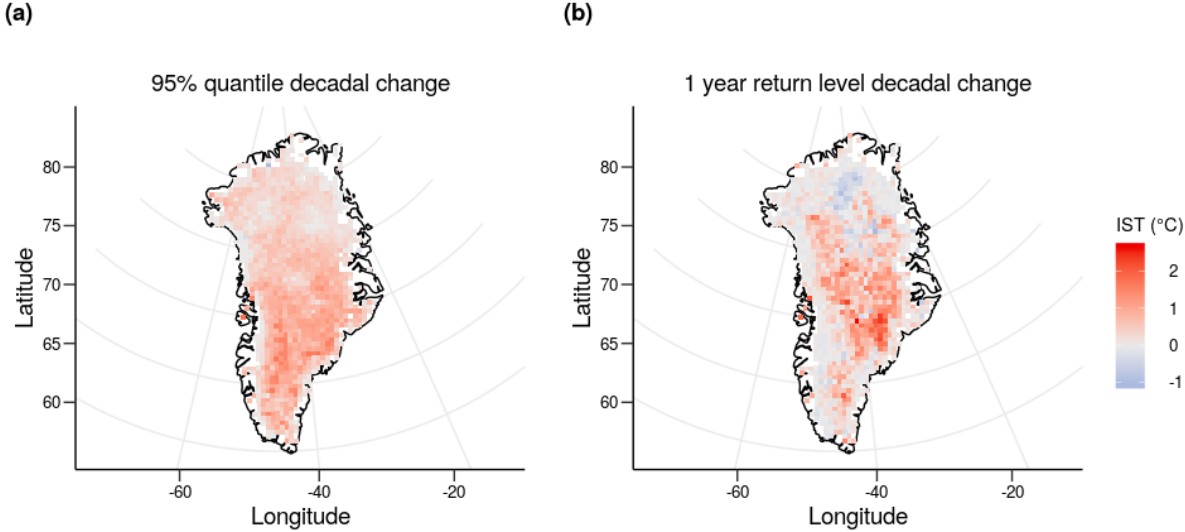

**Figure 8.** Comparison of the change in 95% quantiles (a) and 1 year return levels (b) of mixture models fit to MODIS IST data from 2001 to
2009 and 2010 to 2019.

The 1-year return levels also generally increased, albeit slightly less consistently than the 95% quantiles (849/1139 cells).
Areas in the east show the largest increases - with the largest increase being 2.66°C - however on the south west coast and
particularly the north central area of the ice sheet there are also several cells that show a slight decrease in contrast to larger
increases. More clearly than in the 95% quantiles, 1-year return levels at coastal cells do not increase as much as in central cells
between decades. However, it is important to note that the maxima at coastal cells are already close to or above 0°C. Because
of the soft upper limit of the IST data, values already close to this limit can be partially constrained from further increases, so
cells that had a 1-year return level above 0°C are less likely to show an increase than colder areas such as in the centre of the
ice sheet. This makes the 1-year return level more informative for central cells than for coastal cells.

### 3.4.2 Melt probability

We next compare the probability that each cell experiences melt on any given day for each decade. Using the fitted models, we estimate the probability that each daily observation is a melt temperature, then take an average of all values within our defined decades. For the purposes of interpretation, we limit our discussions to the summer months (May to September, inclusive) when considering melt probabilities, due to the almost zero probability of observing melt outside of this period.

The two decades show very similar trends in their daily melt probabilities, particularly around the coast. However, decade 2 has more cells with a non-zero probability of melt (1017) than decade 1 (853) - an increase of around 19.2% between the two decades - and 68.5% of cells saw an increased probability of melt between the decades. The cell with the single largest probability from either decade is from decade 2 (64.11,-49.93). This cell has a probability of a melt temperature on any given summer day of 0.64 - equivalent to an expected 97.92 melt days per year.

Most of central Greenland has experienced minimal change in the probability of melt between the two decades (Figure 9). This may be largely due to the probabilities being extremely small for these areas regardless of the time period chosen. Coastal cells show clearer and larger cross-decadal variation in melt probability. South east and south west areas of the ice sheet were generally more likely to experience melt, in addition to some cells in the north east and north west areas that were less likely to experience melt, in the more recent decade. The largest increase is on the south east coast, where cells show a maximum change of 0.0351, which equates to an expected increase of 15.42 melt days each year.

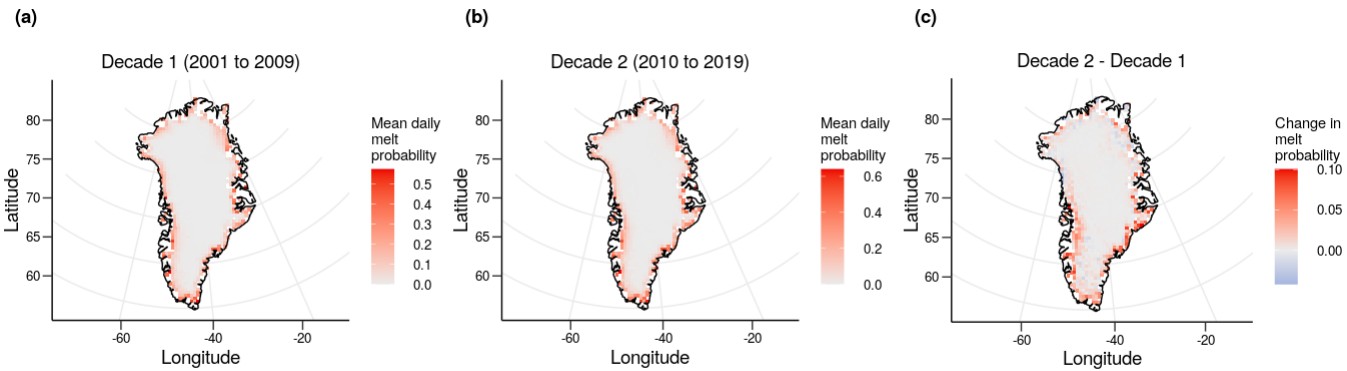

**Figure 9.** The average probability of a melt temperature on any given summer (May-September inclusive) day from 2001 to 2009 (a) and 2010 to 2019 (b), and the change in melt probability between 2001 to 2009 and 2010 to 2019 calculated from the fitted mixture models.

## 4   Discussion

Increases in ice melt in Greenland are of major concern due to the impact that it will have on sea levels (van den Broeke et al., 2016), however in-situ observations of ice melt are sparse, spaced irregularly, and of coarse resolution. Here, we show

that melting can be estimated using a relatively low-dimensional and highly flexible statistical model for IST. This enables us to assemble a record of melting that is continuous in time and space, and is sampled at high spatial (.78 km × .78 km) and temporal (daily) resolution (cloud-cover permitting) using the MODIS IST data set. In addition to the greater availability of IST data, ISTs are on a continuous scale and vary smoothly over time and space, making them better suited to statistical modelling. This is of particular interest given that, from these data, we see that there is ambiguity about whether or not temperatures below 0°C are in fact reflective of melting ice. In this paper, we have addressed this uncertainty by incorporating it into the structure of the statistical model, and thus our record of melting/not-melting is probabilistic rather than binary.

Our model gives comparable results to empirical estimates of melt obtained using a fixed threshold, while also allowing more detailed analyses of melt and the overall temperature distribution via quantile estimation, melt probabilities, and return levels. By modelling the entire temperature distribution, not only can we can gain insight into the frequency and range of melt temperatures, but also broader trends such as higher temperatures in both the high and low quantiles. Furthermore, the model allows for out-of-sample predictions and extrapolation beyond the range of observations. This is of particular interest for melt which occurs with temperatures in the upper tail of the IST distribution where there can in some cells be insufficient data to confidently make empirical estimates.

We observe that melt is much more likely at coastal cells and in the south of the ice sheet than in the centre, and that there is a non-trivial probability of melt occurring below −1°C. The spatial melt trends are in keeping with previous work examining melt using surface mass balance data (van den Broeke et al., 2016) and satellite data (Mernild et al., 2011), including MODIS data (Nghiem et al., 2012). The fitted models also show a clear link between elevation and high ISTs similarly to previous studies linking temperature to elevation (Reeh, 1991), and the yearly expected maxima show the potential for even central areas of the ice sheet to experience melt (Nghiem et al., 2012). Trends previously observed in the south (Mote, 2007) also appeared to have continued, as all cells examined south of 75.16°N saw increases in high-temperature quantiles in the most recent decade.

One of the key considerations is the impact of cloud cover on temperatures, which will not be negligible. The data set used has a complete absence of data on cloudy days. This could be handled in three ways: analyse clear days only, impute missing values, or impute cloudy day data from a second data source. Cloudy day data are not missing at random, since the mechanism which causes the missingness is intrinsically related to the missing values themselves. Consequently the usual methods for imputation using the observed data are not valid. In particular, any such imputation of cloudy day values using the available clear day data would need to take into account the systematic differences between clear and cloudy day temperatures since, as noted, cloudy days are in general warmer than comparable clear days. Because there is a complete absence of cloudy day data, there is no way for the extent of this bias to be estimated empirically. Consequently, we would need to use external information, e.g. other sources of data, to undertake the imputation. This would open up additional problems around different levels of measurement and recording error, different spatio-temporal measurements scales, and so forth, which we believe is beyond the scope of the project.

Although some of the expected annual maxima are just below the lower censoring point of our model's melt component, melt may be possible in these areas over longer time periods. For some cells, the model fit suggests an extremely low probability

of melt. This may be because these cells have few historical instances of possible melt in the data, i.e. no ISTs above the lower censoring point of the melt component. In these instances, the information in the data is insufficient to support a melt component, so only the ice components can be fit to the data, leading to an effective zero probability of melt.

The model also assumes that surface conditions remain similar over the observed time period. Additional impurities becoming present in the ice or rocks appearing after a particularly warm summer could affect the distribution of temperatures at least in the short term and potentially in the long term, however these changes would be difficult to accurately identify using only ISTs data. A separate data set with additional information about surface conditions could be used to identify these changes, or adaptations to the current model structure could be made to allow for the detection of long-term changes in surface conditions. This could take the form a regression or mixed-effects based model, which may represent the surface conditions of the ice but at the expense of being more difficult to fit and potentially interpret due to the increased number of parameters.

Given the assumptions and intuition behind some of the modelling choices, this data set could alternatively be modelled using a Bayesian framework with prior distributions that reflect these assumptions. We would expect melt to have similar distributions at different cells even if there is less evidence of melt in some cells than others. If this is the case, then a modelling framework could be established whereby the melt components of the model share information or parameters, while the ice components are independent between cells. This could be used to estimate melt probabilities even in cells where no melt temperatures have been observed, as melt components could still be estimated using information from other cells with more data resembling melt.

Fitting models to sub-decadal data sets would lead to insufficient data to fit the model; in particular, there would be many cells and time periods with an extremely low number of IST above $-1.65°C$ and $0°C$, making it difficult to fit the melt component with any degree of accuracy. By separately fitting the model to data from two decades (2001 to 2009 and 2010 to 2019), the overall temperature trends were examined. South west and south east areas of the ice sheet were found to have a higher probability of melt in 2010 to 2019 compared to 2001 to 2009, and although $22.2\%$ of cells saw a decrease in melt probability of some degree, $68.5\%$ of cells saw an increase in melt probability and the average increase was more than double the average decrease (-0.0044 compared to 0.011). By contrast, the $95\%$ quantiles increased at almost all cells (1100/1139) and the 1-year return levels increased at most cells (849/1139). The overall trends of the model suggest that melt has become more frequent in the most recent decade, and temperatures more broadly are increasing in areas across the ice sheet.

*Code and data availability.* The MODIS data is available online (https://doi.org/10.5067/7THUWT9NMPDK) from the Multilayer Greenland Ice Surface Temperature, Surface Albedo, and Water Vapor from MODIS, Version 1 dataset (Hall et al., 2018). The AWS station data is available to download online (https://doi.org/10.22008/promice/data/aws) from PROMICE (Fausto et al., 2021). Code for the mixture model and analysis is available upon request at d.clarkson@lancaster.ac.uk.

## Appendix A:  EM algorithm

### A1  Truncated normal distribution

Let $X \sim TN(\mu, \sigma^2, a, b)$ where $\mu$ is the mean, $\sigma$ is the standard deviation, and $a$ ($b$) is the lower (upper) truncation point. Furthermore, let $\alpha = \frac{a-\mu}{\sigma}$ and $\beta = \frac{b-\mu}{\sigma}$. Then $X$ has probability density function:

$$f_{TN}(x) = \frac{f_N(\frac{x-\mu}{\sigma})}{\sigma(F_N(\beta) - F_N(\alpha))},$$

where $f_N$ and $F_N$ are the probability density function and the cumulative distribution function of a standard normal distribution respectively.

### A2  Algorithm

Let $(\mu_k, \sigma_k, \alpha_k, \beta_k)$ denote the parameters for the $k$th truncated normal distribution. To initialise the algorithm, randomly sample without replacement three values of $x \in X$ and set them as $\mu_k$ for $k = 1, 2, 3$. We set $\mu_4 = 0$ to ensure that one of the model components starts in the region of melt temperatures. Let $\sigma_k$ be the sample variance and the component weights $\phi_k = 1/4$ for $k = 1, 2, 3, 4$. For simplicity we refer to the truncated normal probability density function and cumulative distribution function as $f(x)$ and $F(x)$ respectively. The EM algorithm consists of iterating between two stages, the expectation and maximisation steps, until convergence is obtained. For the expectation step, we set:

$$\hat{\gamma}_{ik} = \frac{\hat{\phi}_k f(x_i \mid \hat{\mu}_k, \hat{\sigma}_k)}{\sum_{j=1}^{4} \hat{\phi}_j f(x_i \mid \hat{\mu}_j, \hat{\sigma}_j)}$$

where $\hat{\gamma}_{ik}$ is the estimated probability that observation $i$ belongs to model component $k$.

For the maximisation step, let:

$$\hat{\phi}_k = \sum_{i=1}^{N} \frac{\hat{\gamma}_{ik}}{N}$$

$$\hat{\mu}_k = \frac{\sum_{i=1}^{N} \hat{\gamma}_{ik} x_i}{\sum_{i=1}^{N} \hat{\gamma}_{ik}} + \hat{\sigma}_k \left( \frac{f(\alpha_k) - f(\beta_k)}{F(\beta_k) - F(\alpha_k)} \right)$$

$$\hat{\sigma}_k^2 = \frac{\sum_{i=1}^{N} \hat{\gamma}_{ik}(x_i - \hat{\mu}_k)^2}{\sum_{i=1}^{N} \hat{\gamma}_{ik}} \left[ 1 + \frac{\alpha_k f(\alpha_k) - \beta_k f(\beta_k)}{F_N(\beta) - F_N(\alpha)} - \left( \frac{f(\alpha_k) - f(\beta_k)}{F_N(\beta) - F_N(\alpha)} \right)^2 \right].$$

We iterate between these two steps until the parameters converge to the final estimates (800 iterations was sufficient in this case). The algorithm is considered to have converged if the difference between parameters in each iteration is sufficiently small. We found a difference of $10^{-5}$ between iterations to be sufficient indication of convergence for all parameters.

*Author contributions.*  DC and EE devised the model framework and carried out the data analysis. DC, EE, and AL worked on the interpretation of the model and results and wrote the manuscript.

*Competing interests.* The authors declare that they have no conflict of interest.

*Acknowledgements.* This work was supported by the Data Science for the Natural Environment project (EPSRC grant number EP/R01860X/1).

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
