# Peer review of "Melt probabilities and surface temperature trends on the Greenland ice sheet using a Gaussian mixture model"

_The Cryosphere, 2021_

## Author Response (AR1)

Dear Stef Lhermitte,

We thank you and both reviewers for your comments, insight and the time taken to review our paper. We appreciate the insight and constructive feedback on the paper and for the efforts made to improve it. We have provided detailed responses to the comments made by yourself and both reviewers in this document. If you require any further information, please do not hesitate to contact me at d.clarkson@lancaster.ac.uk.

All the best,

Daniel Clarkson (On behalf of all authors)

**1 Response to Editor's comments**

We thank the editor for their comments and effort taken to review our paper. Their comments were carefully considered and have been incorporated into the improved, revised version of the paper. The original comments made by the editor are in italics, and our responses are in bold. Specific changes to the paper can be seen in the revised version, and any line references made in the below responses relate to the revised version of the paper.

*The role of clouds on the melt/temperatures is neglected completely throughout the manuscript (and only shortly motivated on lines 56-59), but is not expected to be negligible. I think the discussion, conclusion would benefit a lot from a more extensive discussion on the impact, implications etc.*

**We agree with the comments made by the Associate Editor that the impact of cloud cover on temperatures will not be negligible. The data set used has a complete absence of cloudy day values. We see three ways to handle this: analyse clear days only, impute missing values, or impute cloudy day data from a second data source. Cloudy day data are not missing at random, since the mechanism which causes the missingness is intrinsically related to the missing values themselves. Consequently the usual methods for imputation using the observed data are not valid. In particular, any such imputation of cloudy day values using the available clear day data would need to take into account the systematic differences between clear and cloudy day temperatures since, as noted, cloudy days are in general warmer than comparable clear days. Because there is a complete absence of cloudy day data, there is no way for the extent of this bias to be estimated empirically. Consequently, we would need to use external information, e.g. other sources of data, to undertake the imputation. This would open up additional problems around different levels of measurement and recording error, different spatio-temporal measurements scales, and so forth, which we believe is beyond the scope of the project. Instead we have clarified the limitations of the current analysis in the Introduction (lines 46 to 50), and included the above explanation regarding the synthesis of multiple data sets and the consideration of clouds with regards to melt in the Conclusion (lines 248 to 258).**

*The motivation for the multiple ice components is missing (E.g. the multiple blue lines in Figure 2 is missing: where do they come from? How were they calculated etc.?)*

**The multiple blue lines in Figure 2 have been further clarified in the Figure's caption.**

*Especially in the model description: it is not 100% clear to me why you would expect a multi-modal (in this case three) ice temperature components and how they link to the seasons. With a "sinusoidal" temperature distribution, the spring and fall temperatures should be similarish for example. So I think the motivation for a 3-component ice distribution should be shown and motivated way better (e.g. I think the 3-component model in Fig.2 severely overestimates the temperature peaks between*

*-30 and -2C) as you can always fit a model, but you should show that your model is "better" than any other model.*

First we note that the multiple modes are not aiming to reflect seasonality but to capture the shape of the full distribution of temperatures. To show the improvement in the proposed model, and/or to empirically motivate the need for the mixture model, we could fit

**(1) Gaussian model for each season (DJF, MAM, JJA, SON)**

For the simplest solution of fitting a single distribution to the entire dataset, there are 2 separate issues. For cells that experience melt, the mode close to $0°$ C results in at least a bimodal distribution, whilst the majority of cells without melt - that are also comparable to the ice temperature components - have at least two modes even without the mode around $0°$ C. This does not appear to be directly attributable to the seasonal differences in temperatures, as winter, spring and autumn have a variety of distributional shapes at different cells that can't be consistently modelled by a single unimodal distribution. This is also true in summer due to the mode around $0°$ C and a separate mode of the temperatures further below $0°$ C. Fits of unimodal distributions are particularly poor at the tails of the distributions, which is more problematic since our interest lies in the quantity of melt and distribution of melt temperatures which are both concerned with the upper tail of the temperature distribution. An adapted version of this explanation is included in lines 87-93.

**(2) Gaussian model with a sinusoidal mean (and possibly variance)**

The alternative to a mulit-modal distribution would be to model the sinusoidal behaviour directly using a time-series based model, such as a Gaussian distribution with a varying mean and potentially a varying variance. Although this may somewhat capture the seasonal behaviour of temperatures, as with the unimodal models it is less able to accurately model the mode around $0°$ C and the truncation of the temperatures which is where our research interest lies. This further motivates a modelling approach that more directly considers the distribution of the specific data set. Discussion of the possibility of time-series models has been included on lines 102-109.

**(3) Generalised Pareto Distribution**

A further option would be to directly model only the extreme temperatures at the upper tail of the distribution using an extreme value model such as the Generalised Pareto distribution. This would allow the model to focus on the temperatures of highest interest that are the most difficult for more standard models to capture. This also proves problematic though, as in order to fit the model the temperatures must first be identified as extreme using an extreme value threshold, with temperatures above the threshold being classed as extreme and those below being classed as non-extreme. Due to the mode around $0°$ C, finding a consistent threshold location using quantiles, gradient analysis or a specific temperature all encounter problems due to the large variety of tail shapes at different cells. Each of the above mentioned threshold types do not work universally across the ice sheet, and in many cases provide much worse fits than a distribution applied to the whole range of temperatures.

Once a multi-modal distribution was adequately motivated, we chose a Gaussian mixture model due to the seasonal Gaussian model fits having a near normal distribution in some cases. The number of ice components was tested using Bayesian Information Criterion (BIC) for a range of components between from two to seven. Using this metric three ice components was found to be optimal for the largest proportion of cells, so this was chosen as the standard number of ice components for a consistent model across the entire ice sheet. This forms part of the improved Modelling Considerations section 2.2 with a particular focus on lines 87-109.

*The figures contain a lot of unused whitespace (e.g. Figs 4-5), that with a little reorganisation (e.g. removing whitespace edges*

*left/right of GrIS and putting all graphs in one row) can be optimised for eventual publication.*

**Agreed, the figures that were previously in blocks of three split over two rows have been condensed to a single row.**

**2 Response to Referee 1**

80 We thank the reviewer for their time and effort taken to review our paper. Their comments have been carefully considered and accounted for in the revised version of the paper and the individual line corrections have been implemented in the now improved version of the paper. The original comments made by the reviewer are in italics, and our responses are in bold. Specific changes to the paper can be seen in the revised version, and any line references made in the below responses relate to the revised version of the paper.

85 *I think that the authors have done a credible job and that their results are solid. But I also wonder if they have not overcomplicated the analysis. Why not just use the ISTs and IST trends instead of the mixture model? What value added information is provided by the model? I suspect that there is a good reason but it needs to be explained. Please state the reason succinctly in the Abstract, Introduction and Conclusion.*

**The key advantages of a model-based approach over a purely empirical one are that the model permits**

90 **– Estimation of both characteristics of the full distribution of ISTs and certainty in these estimates (i.e. confidence intervals and standard errors). Characteristics include trends, probabilities of exceeding melt threshold(s), quantiles and percentiles, return levels and return periods.**

**– Out-of-sample prediction and extrapolation, again with confidence intervals and/or standard errors. This is particularly important when the characteristics of interest lie in the tail of the distribution where there is typically**
95 **not enough data to be able to make empirical estimates with any degree of confidence.**

**We have added some discussion of this in the Introduction (lines 42 to 44) and Discussion (lines 236 to 239).**

*Also there seems to be some confusion with the words: pixel, location and points. If I am correct, then they all refer to 'cells' in the gridded dataset. Please be consistent or describe the differences.*

**Thank you for noting this; we have replaced all references to 'cell', 'pixel', 'point' or 'location' with 'cell' to be consistent**
100 **with the gridded nature of the data set.**

*Specific comments*

*Line 22 - Instead of the word 'exceeding,' I would say 'equal to or exceeding'*

**Changed**

*23 - you should reference the paper that describes this dataset instead of the reference shown – see below (Hall et al., 2018)*

105 **Thank you, this has been updated**

*24 - this is incorrect; see below for the corrected citation for Wan and Dozier (1989)*

**I have searched for the reference with the exact name of the paper in the corrected reference, however the reference of the paper appeared to be of the same paper as the original reference. The only difference is the order of the first author's names from Zhengming Wan to Wan Zhengming, but this can be re-amended if needed.**

*27 - GCMs should read GCM*

**Changed**

*28 - please add "under clear-sky conditions" after the word 'measurements'*

**Added**

*33 - I would delete 'and future events'*

**Added**

*38 - these are not actually pixels but they are values gridded into cells; therefore, on this line and elsewhere in the manuscript, instead of writing 'pixels' you should write 'cells'*

**Thank you for the clarification, all instances of 'location', 'point' and 'pixel' that refer to cells have been updated to 'cell'.**

*50 - daily should read 'near daily'*

**Changed**

*50 - 'the dataset' should probably read 'this dataset'*

**Changed**

*51 - please check the time span of the dataset; I believe it ends on 12/31/2020 andnot in 2019*

**This has been re-checked and the current time span is correct as of the dataset description at https://nsidc.org/data/MODGRNLD/versions/1.**

*54 I would replace the word 'points' with 'cells'*

**This and other occurrences of points have been replaced with cells for consistency**

*58 delete the words 'a slight'*

**Changed**

*61 add 'because of cloud cover' after 'summer months'*

**Changed**

*Fig 1 caption: replace the words 'non melting' with 'available;' instead of 'locations,' say 'cells'*

**'Non-missing' has been updated to 'available', and 'locations' has been updated to 'cells' both here and in any other locations in the paper for consistency.**

*72 delete single parenthesis*

**Changed**

*99 please check original reference; this should depend on percent salinity*

**This is correct that the melting point depends on the percent salinity of the ice. By taking the value used in the referenced paper as the lower limit for the melting point, we aim to take a conservative lower bound on the melting point rather than a precisely accurate value to ensure that all possible melt temperatures are considered in the density of the melt component. The specific text of note in the reference is: "Because sea ice is saline, it freezes at a temperature that is less than 273.15 K (the freezing point of fresh water); we use 271.5 K as the cutoff temperature between water and ice for the "cold-period" images in the Arctic Basin, although a user can select their own cutoff temperature based on the IST to develop a sea ice extent map."**

*121 there is also the +/- 1 deg C accuracy to consider*

**The +/- 1°C accuracy is not directly considered in terms of the explicit value of the accuracy, but it is considered from the perspective of the model structure and in terms of classifying observations as melt temperatures. The typical methodology of using -1 °C as the melt threshold only considers the accuracy in that it is the melting point of ice minus the accuracy. In our model structure, the overlap region between the ice and melt components allows for melt temperatures as low as -1.65 °C to be modelled as melt, allowing for some inaccuracy in the observations to be accounted for when deciding whether they are ice or melt temperatures. We tested directly fitting the truncation points in an earlier version of the model, however the flexibility of the model structure and the increased number of parameters this created made them difficult to fit in practice. We also previously tested the model's sensitivity to changing the truncation points by re-fitting the model with marginally different truncation points, but this made only minor differences to the overall parameters when changing the lower truncation point which was the more unknown of the two truncation points prior to modelling the data.**

*130 - 'cell.' is 'sample' the same as 'point,' 'pixel' or cell? If yes, please be consistent and use*

**This and all other instances of 'sample' and 'pixel' have been changed to 'cell' for consistency.**

*134 - do you mean greater than or equal to -1 deg C?*

**Changed**

*Figure 4 in (a) do you mean greater than or equal to -1 deg C? The heading should say* 'observed.'

**This comment is correct, the legend and heading have both been corrected.**

*138 - since you are referring to the glacier facies/zones, you should cite Benson (1962).*

**Thank you for correcting this, a reference to Benson has been added.**

*151, 159, 172 173 and elsewhere use 'cells.'*

**This has been updated in line with other changes to terminology to the consistent phrase of 'cells'**

*192-193 - are 'locations' the same as cells? If so, then please is this a statistically significant difference?*

**As above, this has been changed to 'cells'**

*203 please delete the word 'etc'*

**Changed**

*206 the spatial resolution of the product is 0.78 X 0.78 km, not 1 km as stated*

**Thank you for the correction, this has been changed.**

*206 after the word 'resolution,' please add the following "(cloud-cover permitting)"*

**Added**

**3    Response to Referee 2**

We thank the reviewer for their time and effort taken to review our paper. Their comments have been carefully considered and accounted for in the revised version of the paper. We addressed their constructive comments regarding the interest and aims of the research and believe we have improved the paper as a result. The original comments made by the reviewer are in italics, and our responses are in bold. Specific changes to the paper can be seen in the revised version, and any line references made in the below responses relate to the revised version of the paper.

*This short paper presents an innovative method to derive melt extent from MODIS surface temperature (ST) based on its statistical distribution. While the presented work seems to be statistically robust, I don't see the interest of this complex method with respect to a simple one based on TS > 0°C.*

**Whilst interest is primarily in understanding and predicting positive temperature events, there are several reasons for using more than just the positive observations. In the first instance, for many sites, the number of positive values is insufficient for us to be able to fit a model (or indeed estimate their properties) with any degree of confidence. Secondly, for many sites there is an interest in learning about the behaviour of sub-zero temperatures to ascertain whether there is an upwards shift in the overall distribution which could, if it continues, impact the frequency and magnitude of positive temperature events.**

*If TS values larger than 0°C are sometimes retrieved in MODIS, it is due to the presence of rocks or impurities at the surface of the ice sheet (ST > 0°C can also been seens in the PROMICE AWSs data set).*

[Figure]

**Figure 1.** Indicator of whether any ISTs at each cell in our sample are greater than or equal to 0 $^\circ$ C.

**This may be the case, but if so we would expect to see the positive values to be isolated in space. In fact, inspection of the data shows that ISTs greater than or equal to $0^\circ$ C are seen much more commonly and are more widespread than would be expected from only rocks or impurities (See Figure 1). In particular, cells near the coast that would be expected to experience melt do not have any ISTs $\geq 0^\circ$ C, suggesting that this cannot be used as the melt threshold. Similarly, cells much further inland that would be expected to have a lower probability of impurities or exposed rock have ISTs $\geq 0^\circ$ C, suggesting that values above $0^\circ$ C are not purely for either of these features.**

*Therefore a comparison with a melt extent independent data set (like a microwave product) is needed to show the interest of this new method.*

**The purpose of the proposed methodology is not solely to produce a binary melt extent data set, but to improve understanding of the full distribution of temperatures and particularly those close to the melt threshold.**

*Moreover, to build the statistic model, several years of observations are needed by assuming that the surface conditions remain similar. But what occurs, if rock or impurities appear in surface after a summer with a huge melt? Are the algorithm enough robust to deal with such problem? Idem, what about percolation zone pixels becoming bare ice zone pixels through the recent summers? I suspect that the statistic distribution of these pixels could change... More details about these potential issues are also needed for me.*

**Indeed the model as presented requires the assumption that the surface conditions remain similar over the period of time for which data are collected. We have added a comment on this limitation to the Discussion (lines 264 - 270) The**

195

200

205

model could be extended to allow for long-term or inter-year changes in the ice sheet surface. The most straightforward way to do this would be through either a regression- or mixed-effects modelling framework. This would involve modelling all, or a subset, of the parameters of the mixture model (component weights, means and variances) as functions of time, annual random effects, or any other physical covariates that are expected to affect the surface temperature. Such a model could be highly flexible, with appropriate forms for the parameters chosen using a statistical model selection process. The disadvantages of this kind of model are that it would be harder to fit (more parameters to estimate) and model predictions are conditional on time/year/covariates and thus harder to interpret, though arguably more realistic.

---

## Author Response (AR2)

Dear Stef Lhermitte,

We thank you and both reviewers for your second round of comments, and further time spent reviewing our paper. We appreciate the additional constructive feedback on the paper made in response to our initial attempts to respond to your points. We have provided detailed responses to the comments made by yourself and reviewer 2 in this document. If you require any further information, please do not hesitate to contact me at d.clarkson@lancaster.ac.uk.

All the best,

Daniel Clarkson (On behalf of all authors)

**1    Response to Editor and Reviewer's comments**

We thank the editor and both reviewers for their comments and effort taken to review our paper. Their comments were carefully considered and have been incorporated into the improved, revised version of the paper. We have addressed only the reviewer's comments below since the editor's comments are a summarised version of the same points. The original comments made by the reviewer are in italics, and our responses are in bold. Specific changes to the paper can be seen in the revised version, and any line references made in the below responses relate to the revised version of the paper.

*Thanks for this revised version and your responses. But I'm continuing to think that a comparison with a microwave based melt product is needed, although I agree that your product is well more than a simple binary melt extent data set. It is nice to show and discuss melt and temperature evolution but we don't know how your product performs. Is it reliable and robust? Is it able to detect the extreme melt events (mostly driven by clouds) knowing that only the cloud-free days are considered here? In the accumulation zone, melt occurs when there are clouds and in the ablation zone, only  50% of days are cloud free.*

*As cloudiness has been changing and is different between the 2 decades 2001-2009 and 2010-2019, can these 2 decades be compared in Section 3.3 if you consider only the cloud-free days? How do your temperature statistics compare with the ones using surface temperate from ERA5 for example ?*

**We first note that our choice of the Moderate Resolution Imaging Spectroradiometer (MODIS) data was for (a) high resolution spatial coverage and (b) the availability of a full range of ice surface temperature measurements. The latter is of particular importance as it allows us not just to investigate melt itself, but also to study the behaviour of those temperatures in the range just below the melt threshold and to investigate these for trends and patterns. An unfortunate limitation of using MODIS is that our results are constrained to inferences under clear sky conditions only. Notwithstanding this, we can still make useful inference about long term temperature and melt trends under these conditions. To address the reviewer's concern, we have added a further investigation looking for evidence of changes in cloud cover both spatially and over time. We found minimal inter-year cloud cover variability between the two periods for which clear-sky day temperatures are subsequently compared: 2001 to 2009 and 2010 to 2019, periods which were in any case originally chosen to limit the impact of inter-year variability of climate and weather on any trends observed. In particular, differences in cloud cover over time are negligible for those months with the greatest proportion of melt, providing further confidence that weather conditions/patterns have not significantly changed over the range of the data and thereby that surface conditions are sufficiently similar over time to be modelled by a single distribution.**

*Therefore, you need to show, at least over one season (e.g. summer 2019) what is the interest and robustness of your complex model with respect to a microwave data set (Can your model be applied to the microwave brightness temperatures?). An other validation, if your prefer, would be to compare the melt at the PROMICE AWS's using measured temperatures as the "true"*

*melt data set.*

A direct comparison between either the Automatic Weather Station (AWS) and MODIS data or between the fit of our model to the two sets of data is severely limited by a fundamental difference in the characteristics between the two datasets: namely that MODIS measurements contain a non-negligible number of positive values, and AWS observations do not. Indeed, it is precisely this characteristic of the data that informed the final form of our proposed model, since the mass of observations close to and just above $0°$C meant that a routine statistical model for the upper tail of the data was inappropriate. Since the AWS does not display this trait, we would not expect the proposed model to fit as well. Adaptation, such as removing the fourth 'melt' component from the mixture distribution, and replacing it with a right-censored third component, could be made but are beyond the scope of the paper. We would also like to point out that a primary objective of our model is to describe and predict the full temperature distribution at each sample location, with estimation of melt extent following as a secondary consequence.

Nevertheless, as recommended by the reviewer, we have provided an additional subsection in the paper comparing the melt found by applying our ice surface temperature model to MODIS Ice Surface Temperature (IST) data with that of the Programme for Monitoring of the Greenland Ice Sheet (PROMICE) AWS data. Given the good fit of our model to the MODIS dataset, this acts, to a high degree of approximation, as a comparison of the melt observed at the AWSs and that captured by MODIS as opposed to a comparison of the former with the melt estimates from our fitted model. We further reference previous studies comparing the accuracy of MODIS and AWS data sets to establish a foundation for the discussion, and compare the estimated melt at all PROMICE AWSs to the closest of our sub-sampled MODIS cells. These comparisons must be considered with caution since the distances between the individual AWSs and their nearest neighbour MODIS cell ranges from 4–59 km.

*Finally, with respect to Casey et al. (2017), how could your product be impacted by the changes of sensors in the MODIS data set knowing that your model needs no change in the surface conditions?*

*Ref: Casey, K. A., Polashenski, C. M., Chen, J., and Tedesco, M.: Impact of MODIS sensor calibration updates on Greenland Ice Sheet surface reflectance and albedo trends, The Cryosphere, 11, 1781–1795, https://doi.org/10.5194/tc-11-1781-2017, 2017.*

The paper referred to makes note of sensor calibration updates with respect to surface reflectance and albedo, but not for surface temperature. The paper examines data from MODIS bands 1 to 7, whereas ice surface temperature is derived using bands 31 and 32 (https://nsidc.org/sites/nsidc.org/files/technical-references/MOD29_C61_UserGuide.pdf).

---

## Author Response (AR3)

Dear Stef Lhermitte,

We thank you and both reviewers for the further time spent reviewing our paper and for accepting it for final publication in The Cryosphere. We are grateful for the feedback given throughout the review process and for the improvements that have been made to the paper as a result.

All the best,

Daniel Clarkson (On behalf of all authors)